# Visual Health and Academic Performance in School-Aged Children

**DOI:** 10.3390/ijerph17072346

**Published:** 2020-03-31

**Authors:** Cristina Alvarez-Peregrina, Miguel Ángel Sánchez-Tena, Cristina Andreu-Vázquez, Cesar Villa-Collar

**Affiliations:** Universidad Europea de Madrid-School of Biomedical and Health Science, 28670 Madrid, Spain; miguelangel.sanchez@universidadeuropea.es (M.Á.S.-T.); cristina.andreu@universidadeuropea.es (C.A.-V.); villacollarc@gmail.com (C.V.-C.)

**Keywords:** vision, academic performance, school-aged children, health, growth

## Abstract

Background: Academic performance at different educational levels has become a very important subject of study in local, national and international institutions. A visual system working properly is critical to improving academic performance. It is important to check children’s vision during the different stages of elementary school. Methods: A visual screening was carried out in elementary school children, aged between 6 and 12, across Spain. The screening included: the collection of demographic data, surveys of children and families about their vision habits and a basic optometric screening to detect visual problems. Results: Children with bad academic performance had worse visual health than those with good academic performance. Conclusions: It would be highly recommendable to introduce policies that ensure the early detection of visual disorders in schools and primary care in order to improve the academic performance of elementary students.

## 1. Introduction

Nowadays, academic performance at different educational levels has become a very important subject of study in local, national and international institutions. The OECD’s Program for International Student Assessment (PISA report 2019) [1,2,3], which has been carried out annually since 2000, has a great impact, and makes the governments of the different studied propose educational policies to improve the performance of their students.

There are different methods of measuring academic performance in children, such as standardized achievement test scores [4], teacher ratings of academic performance [5], and report card grades. In Spanish elementary schools, academic performance is measured by grades as follows: *Insuficiente* (IN) for negative qualifications (bad academic performance) and *Suficiente* (SU), *Bien* (BI), *Notable* (NT), or *Sobresaliente* (SB) for positive qualifications (good academic performance) [6].

Academic performance is conditioned by many factors; some of them come from the context and others from the person [7]. This makes it necessary that governments propose policies that help to improve all of the aspects that can affect academic performance, rather than simply making changes in syllabi or learning methods.

Children’s health affects their academic performance, influencing the educational aspiration of adolescents. These aspirations are lower in students that have suffered from health problems [8].

Visual health can have a great influence on students’ academic performance. Shin et al. have already highlighted that visual disorders are one of the best predictors of academic performance [9]. However, there is little evidence on the impact of vision on academic performance in children in elementary school. In Australia, White, Wood, Black and Hopkins concluded, in a series of studies of different grades of elementary school, that better visual information processing results in better academic performance [10,11,12]. In Malaysia, in 2011, Chen et al. studied 1103 grade 2 school children, supporting the concept that visual performance is key to learning [13].

In Spain, well-child visits by pediatricians include vision checks at ages that vary among the different Autonomous Communities. According to the recommendations given in the guide of preventive activities by age group, from the PrevInfad Group and PAPPS Childhood and Adolescence, a complete visual screening should be conducted for children aged between 2 and 4 [14]. However, there is a lack of scientific literature in Spain that supports the relationship between vision and learning.

As previously mentioned, a visual system properly working is critical to improve academic performance. It is therefore important to check children´s vision during the different stages of elementary school.

## 2. Materials and Methods

A visual screening was carried out on elementary school children, aged between 6 and 12, all over Spain. The screening included:-The collection of demographic data, including sex and age.-A survey of children about their vision in daily life situations.-A survey of families about their children’s visual habits.-A basic optometric screening to detect visual problems, including:oMonocular visual acuity in far visionoBinocular visual acuity in near visionoNear point of convergenceoOcular motility

The screening was done by optometrists in optical centers that collaborated in the study ‘State of children’s vision in Spain 2019’, organized by the *Vision y Vida* Foundation. Each optometrist had a username and password to upload the collected data on the website www.verparaaprender.es.

For all participants and each of the study cohorts, qualitative variables were described using absolute and relative values. Quantitative variables were described using means ± standard deviation (SD), or medians and interquartile range [IQR], depending on their distribution (Shapiro-Wilk normality test). Chi-square tests and Student T-tests from independent samples or Mann Whitney U tests were used to compare study cohorts. Odds ratios and 95% confidence intervals were calculated for qualitative variables. Differences between the means of participants with “BAD” and “GOOD” academic performance, and 95% confidence intervals, were calculated for quantitative variables. The analyses were performed using Stata IC v.14 (StataCorp), and a significance level of 5% was established.

The research described herein adheres to the tenets of the Declaration of Helsinki, and was approved by the ethics investigation committee of the *Universidad Europea de Madrid* (CEI-UE). All records were anonymous; only statistical information was provided by *Fundacion Visión y Vida* for research purposes.

## 3. Results

In total, 11406 children participated in the visual screening. 1188 children were excluded as the surveys did not contain information about their academic performance. The 10,218 children included in the analysis were classified according to their academic performance. The sample was divided into 1006 children (9.85%) with bad academic performance and 9212 (90.15%) children with good academic performance.

### 3.1. Demographical Data

47.96% of the children were male, while 52.04% were female. The proportion of males was higher in the group with bad academic performance than in the group with good academic performance (57.21% versus 46.95%; *p* < 0.001).

The age of the participants was 8.08 ± 2.04 years old. The median was 8 [4] years old in all the sample. Table 1 shows the detail of demographical data.

### 3.2. Survey to Children about Their Vision in Daily Life Situations

Table 2 shows the proportion of children that answered ‘yes’ or ‘no’ to the questions about their habits related to vision. There were differences in the proportion of subjects answering ‘yes’ to each question between the cohorts with bad and good academic performance.

### 3.3. Survey of Families about Children’s Visual Habits.

Table 3 shows the proportion of families that answered ‘yes’ or ‘no’ to the questions about their children’s visual habits.

Families of children wearing glasses were also asked about how they noticed that their children needed glasses. Most of the families said that it was the child who let them know (8.94% of the sample, 8.91% of children with good academic performance and 10.14% of the children with bad academic performance). The second most reported reason was a visit to the optometrist or ophthalmologist (4.26% of the subjects, 5.77% in the cohort with bad academic performance and 4.09% in the cohort with good academic performance). The third most reported reason was s parent’s observation (3.17% of the subjects, 4.37% versus 3.04% in the cohort with good and bad academic performance respectively). In 234 subjects (3.69%) it was noticed differently.

### 3.4. Visual Screening

Table 4 shows values of monocular visual acuity in far vision, binocular visual acuity in near vision and near point of convergence. It shows data from the total of the subjects, and of both cohorts. There are differences between both cohorts for all of the values measured (*p* < 0.001).

Regarding binocular vision, 8.88% of children had disorders. The percentage increased in the cohort with bad academic performance (19.98%) and decreased in the cohort with good academic performance (7.66%) (*p* < 0.001)

Finally, ocular motility was checked, with 88.67% of children with normal movements. The percentage decreased to 76.64% of children in the group with bad academic performance, and increased to 89.98% of the children with good academic performance (*p* < 0.001).

## 4. Discussion

It should be noted that the consistency of this study is due to the large sample studied, numbering more than 10000 children. In order to classify the sample in both cohorts, good and bad academic performance, this study has been based on the families’ answers. The percentage of children with bad academic performance is aligned with the results of elementary education that the Spanish Ministry of Education and Vocational Training published [15].

Regarding gender, we have found more males than females within the bad academic performance group. This data is aligned with the OCDE’s reports about gender equality in education. According to these reports, at age 15, 60% of the lowest achievers in mathematics, reading, and science are boys, whilst 40% are girls [16].

Analyzing the children’s answers about their habits related to vision, there are differences between children with bad academic performance and those who have a good academic performance, with vision being better in the good academic performance group. There is a higher percentage of children that see the blackboard properly, and who consider themselves to have good distance and near vision, within the cohort with a good academic performance. Also, the percentage of children that get tired while reading, suffer from headaches, feel that their eyes cry or itch when they read, need to follow the text with their finger, or even get confused or miss words when reading, is lower in the group of children with good academic performance. These results are aligned with other studies carried out in different countries, such as that published by Jan et al., which concluded that better visual acuity is related to higher academic performance in a cohort of children aged between 11 and 16 in China [17]. Regarding reading, Goldstand et al. demonstrated in 2006, in 71 seventh grade children, how nonproficient readers had significantly poorer academic performance and vision-screening scores than proficient readers [18].

By the analysis of the family’s answers, it was found that a higher percentage of the families of the group of children with bad academic performance think that their children do not see properly, get close to TV or paper, get up frequently when they do their homework, and that their children’s eyes become red after a long visual effort. Despite some studies showing how wearing corrective lenses improves children’s wellbeing and school function [19,20], in this cohort of children with bad academic performance more children do not like reading and also wear glasses or contact lenses.

Results from visual screening show that visual acuity in distance vision is lower in children with bad academic performance, and that they also have a longer near point of convergence. Regarding binocular vision, there are more children with problems with fusion images with both eyes within the cohort with bad academic performance. These data are also aligned with other studies such as Hopkins’ review, which concluded that there is a concordance between academic performance and visual acuity and refractive error [21].

There are several studies that propose different methods of vision screening in schoolchildren. Indeed, some associations have published their recommendations, such as ‘Procedures for the Evaluation of the Visual System by Pediatricians’ written by Donahue et al. in 2016 [22], or the guidelines published by the Expert Committee of the Canadian Association of Optometrists and the Canadian Ophthalmological Society [23]. However, there are no universally agreed upon policies or strategies for vision screening in children. According to Hopkins et al., this is due to the lack of evidence supporting the benefits of screening, and to the lack of support of the authorities [24]. Our study re-confirms the importance of vision in children, using a large sample of them checked by optometrists.

It is also important to highlight the way in which families with children wearing glasses realized that they needed them. In most of the cases, the children themselves realized that they did not see properly. The second and third most reported options are prescription by a specialist and warnings from schools. These remarkable results, and the fact that visual factors are better predictors of academic success than others [25], make it highly recommendable to establish policies in primary care and schools that diagnose visual problems as soon as possible, in order to improve children’s academic performance.

## 5. Conclusions

Visual health is better in children with good academic performance than in those with bad academic performance.

It would be highly recommendable to introduce policies that assure the early detection of visual disorders in schools and primary care, in order to improve the academic performance of elementary students.

## Figures and Tables

**Table 1 ijerph-17-02346-t001:** Demographic analysis of the study population.

	TOTAL	BAD Academic Performance	GOOD Academic Performance	*p*-Value
**N of participants**	10218	1006	9212	
**(% of total)**		(9.85%)	(90.15%)	
**Age (years old)**				
**Mean ± SD**	8.08 ± 2.04	8.43 ± 2.10	8.04 ± 2.03	<0.001
**Median [IQR]**	8.00 [4.00]	8.00 [3.00]	8.00 [4.00]	
**N of female participants ***	5311	430	4881	
**(% of total participants with known gender)**	(52.04%)	(4.21%)	(47.83%)	<0.001
**(% of cohort of study with known gender)**		(42.79%)	(53.05%)	

* The answers “NR/DK” are not included.

**Table 2 ijerph-17-02346-t002:** Answers given by children about their vision in daily life situations.

	TOTAL		Bad Academic Performance		Good Academic Performance		*p*-Value	OR (95% CI) **
	No	Yes	No	Yes	No	Yes		
**Do you see the blackboard properly? ***	1323 (12.95%)	8507 (83.26%)	246 (24.45%)	696 (69.18%)	1077 (11.69%)	7811 (84.79%)	**<0.001**	**0.39** **(0.33–0.46)**
**Do you have a good far and near vision? ***	1626 (15.91%)	7990 (78.20%)	307 (30.52%)	593 (58.95%)	1319 (14.32%)	7397 (80.30%)	**<0.001**	**0.34** **(0.30–0.40)**
**Do you get tired while reading? ***	6985 (68.36%)	2537 (24.83%)	486 (48.31%)	427 (42.45%)	6499 (70.55%)	2110 (22.90%)	**<0.001**	**2.71** **(2.35–3.12)**
**Do you suffer from headaches or your eyes cry or itch while reading? ***	7454 (72.95%)	2159 (21.13%)	577 (57.36%)	353 (35.09%)	6877 (74.65%)	1806 (19.60%)	**<0.001**	**2.33** **(2.02–2.69)**
**Do you have anytime double vision? * **	8426 (82.46%)	1067 (10.44%)	738 (73.36%)	164 (16.30%)	7688 (83.46%)	903 (9.80%)	**<0.001**	**1.89** **(1.57–2.28)**
**Do you need to follow the text with your finger when you read? ***	6195 (60.63%)	3369 (32.97%)	458 (45.53%)	469 (46.62%)	5737 (62.28%)	2900 (31.48%)	**<0.001**	**2.03** **(1.76–2.33)**
**Do you get confused or miss words when reading? ***	6212 (60.79%)	3072 (30.06%)	390 (38.77%)	521 (51.79%)	5822 (63.20%)	2551 (27.69%)	**0.014**	**3.05** **(2.65–3.51)**

* The answers ‘NR/DK’ are not included; ** The odds ratio (and its 95% confidence interval) of answering ‘yes’ to each of the questions for participants with ‘bad academic performance’, with respect to participants with ‘good academic performance’. Significant *p*-values and OR (*p* < 0.05) are shown in bold.

**Table 3 ijerph-17-02346-t003:** Answers given by families about their children’s visual habits.

	TOTAL		Bad Academic Performance		Good Academic Performance		*p*-Value	OR (95% CI) **
	No	Yes	No	Yes	No	Yes		
**Do you think that your child has good vision? ***	1315 (12.87%)	8143 (79.69%)	292 (29.03%)	597 (59.34%)	1023 (11.11%)	7546 (81.91%)	**<0.001**	**0.28** **(0.24–0.32)**
**When your child is watching TV, does he/she get close to the screen or squint his/her eyes? ***	7667 (75.03%)	2230 (21.82%)	584 (58.05%)	375 (37.28%)	7083 (76.89%)	1855 (20.14%)	**<0.001**	**2.45** **(2.13–2.83)**
**When your child is reading, does he/she get close to the paper? ***	6860 (67.14%)	2928 (28.66%)	450 (44.73%)	491 (48.81%)	6410 (69.58%)	2437 (26.45%)	**<0.001**	**2.87** **(2.50–3.30)**
**Does your child get up frequently when he/she does his/her homework? ***	5722 (56.00%)	3960 (38.76%)	269 (26.74%)	680 (67.59%)	5453 (59.19%)	3280 (35.61%)	**<0.001**	**4.20** **(3.62–4.89)**
**Are your child’s eyes red after a long visual effort (TV, homework, etc.)? ***	7900 (77.31%)	1684 (16.48%)	602 (59.84%)	322 (32.01%)	7298 (79.22%)	1362 (14.79%)	**<0.001**	**2.87** **(2.47–3.33)**
**Does your child like to read? ***	3142 (30.75%)	6592 (64.51%)	683 (67.89%)	282 (28.03%)	2459 (26.69%)	6310 (68.50%)	**<0.001**	**0.16** **(0.14–0.19)**
**Does your child wear glasses?**	7974 (78.04%)	1237 (12.11%)	735 (73.06%)	157 (15.61%)	7239 (78.58%)	1080 (11.72%)	**<0.001**	**1.43** **(1.19–1.73)**
**Does your child wear contact lenses?**	7009 (68.59%)	59 (0.58%)	636 (63.22%)	14 (1.39%)	6373 (69.18%)	45 (0.49%)	**<0.001**	**3.12** **(1.57–5.82)**
**Does your child suffer from myopia? ***	6643 (65.01%)	425 (4.16%)	604 (60.04%)	46 (4.57%)	6039 (65.56%)	379 (4.11%)	0.231	1.21 (0.86–1.67)
**If your child is wearing glasses, does he/she like to wear them? *****	267 (21.58%)	843 (68.15%)	45 (28.66%)	94 (59.87%)	222 (20.56%)	749 (69.35%)	**0.014**	**0.62** **(0.42–0.93)**

* The answers ‘NR/DK’ are not included; ** The odds ratio (and its 95% confidence interval) of answering ‘yes’ to each of the questions for participants with ‘BAD academic performance’, with respect to participants with ‘GOOD academic performance’. Significant *p*-values and OR (*p* < 0.05) are shown in bold. *** Only the answers of children wearing glasses are included.

**Table 4 ijerph-17-02346-t004:** Results from visual screening.

	TOTAL			BAD Academic Performance			GOOD Academic Performance			*p*-Value	Difference ± SE (95% CI) *
	n	Mean ± SD	Median [IQR]	n	Mean ± SD	Median [IQR]	n	Mean ± SD	Median [IQR]		
**Monocular VA (RE)**	10192	0.92 ± 0.15	1.00 [0.10]	1000	0.88 ± 0.18	1.00 [0.20]	9192	0.93 ± 0.14	1.00 [0.10]	**<0.001**	**−0.05 ± 0.00** **(−0.06−–0.04)**
**Monocular VA (LE)**	10201	0.90 ± 0.16	1.00 [0.10]	1003	0.84 ± 0.20	0.90 [0.20]	9198	0.91 ± 0.16	1.00 [0.10]	**<0.001**	**−0.07 ± 0.01** **(−0.08−–0.06)**
**Binocular VA near distance**	7042	0.97 ± 0.09	1.00 [0.00]	650	0.94 ± 0.13	1.00 [0.10]	6392	0.97 ± 0.09	1.00 [0.00]	**<0.001**	**−0.03 ± 0.00** **(−0.04–−0.03)**
**NPC (cm)**	9729	5.51 ± 5.55	5.00 [7.00]	934	6.47 ± 6.08	5.00 [9.00]	8795	5.40 ± 5.49	5.00 [7.00]	**<0.001**	**1.06 ± 0.19** **(0.69–1.44)**

* Difference ± standard error (and its 95% confidence interval) calculated as the mean of participants with “bad academic performance”, minus the mean of participants with “good academic performance”. Significant differences (*p* < 0.05) are shown in bold.

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
