# Peer review of "Visual Health and Academic Performance in School-Aged Children"

_ijerph, 2020, doi:10.3390/ijerph17072346_

Round 1

Reviewer 1 Report

I consider this MS to have valuable data that would be of interest, however it needs a major revision before further consideration. Major concerns are: 1. The abstract should have a proper structure i.e. background, material and methods, results and conclusions. 2. The authors should add demographic analysis of the study population in a separate table 3. The authors should add the statistical analyses to the tables and consider more sophisticated risk factor analysis (OR) 4. There is a lack of ethical statement  5. Last but not least a discussion of the methods of vision screening in schoolchildren would be a valuable addition to this MS

Author Response

We are very excited to have been allowed to revise our manuscript. We carefully considered your comments.

In the Word file we explain how we revised the paper based on those comments and recommendations. We want to extend our appreciation for taking the time and effort necessary to provide such insightful guidance.

Reviewer 2 Report

Visual health is critical topic and it deserves a proper intervention in order to promote good academic performance. However, this conclusion it is not new at all. There is a good amount of literature that pointed this fact.  This study shows that visual health is better in children with good academic performance and authors suggest to introduce policies that assure the detection of visual impairments.  Authors analyze a good number of elementary school students (10218) and observed that males showed a higher incidence of bad performance and vision problems, but they did not discuss about this observation.

In my opinion this data is solid, but it does not provide enough information to be published as an academic paper in IJERPH. It lacks novelty and the discussion is very poor.  Also, there is a number of references about similar studies that should be included/compared in the discussion and also in the introduction.  

Statistical differences between groups should be included in the tables. 

Lane 85: “There were differences between the proportion of the answers in the cohort with bad and Good academic performance in all the variables”. This is poorly described.

Lane 79 and 86: “Good” should not include capital letter “G”

Author Response

We appreciate for taking the time and effort necessary to provide such insightful guidance. We hope you can reconsider the article after all the changes we have done.

In the Word file we explain how we have revised the paper based on your comments and recommendations. 

Reviewer 3 Report

Thank you for this interesting manuscript.

The results are important, but I have two queries. I lack information about the definition of good academic performance and bad academic performance. The conclusions are based on these definitions, and it is important to know the definitions.

The study is based on large groups, and this gives a risk of type-1 error. Please, describe your thoughts about this.

Author Response

Thank to you for your time in reviewing this paper. We have carefully considered your comments.

Following (in the word file) we explain point by point how we have revised the paper based on those comments and recommendations.

Round 2

Reviewer 1 Report

I am satisfied with the changes made by Authors

Reviewer 2 Report

Authors improved this manuscript based in reviewers comments and suggestions. 

Reviewer 3 Report

I think that the authors have given response on the queries, and that the manuscript now is suitable for publication.